# Cyclin E1 in Murine and Human Liver Cancer: A Promising Target for Therapeutic Intervention during Tumour Progression

**DOI:** 10.3390/cancers13225680

**Published:** 2021-11-13

**Authors:** Roland Sonntag, Christian Penners, Marlene Kohlhepp, Ute Haas, Daniela Lambertz, Andreas Kroh, Thorsten Cramer, Fabio Ticconi, Ivan G. Costa, Frank Tacke, Nikolaus Gassler, Christian Trautwein, Christian Liedtke

**Affiliations:** 1Department of Internal Medicine III, University Hospital RWTH, 52074 Aachen, Germany; cpenners@ukaachen.de (C.P.); uh57@arcor.de (U.H.); dlambertz@ukaachen.de (D.L.); ctrautwein@ukaachen.de (C.T.); 2Department of Hepatology and Gastroenterology, Campus Virchow-Klinikum and Campus Charité Mitte, Charité University Medicine Berlin, 13353 Berlin, Germany; marlene.kohlhepp@charite.de (M.K.); frank.tacke@charite.de (F.T.); 3Department of General, Visceral and Transplantation Surgery, University Hospital RWTH, 52074 Aachen, Germany; akroh@ukaachen.de (A.K.); tcramer@ukaachen.de (T.C.); 4Department of Surgery, Maastricht University Medical Center, 6200 MD Maastricht, The Netherlands; 5ESCAM—European Surgery Center Aachen Maastricht, 52074 Aachen, Germany; 6ESCAM—European Surgery Center Aachen Maastricht, 6200 MD Maastricht, The Netherlands; 7IZKF Research Group Computational Biology and Bioinformatics, Helmholtz Institute for Biomedical Engineering, RWTH Aachen University, 52074 Aachen, Germany; fabio.ticconi@rwth-aachen.de (F.T.); ivan.costa@rwth-aachen.de (I.G.C.); 8Section of Pathology, Institute of Forensic Medicine University Hospital Jena, 07747 Jena, Germany; nikolaus.gassler@med.uni-jena.de

**Keywords:** cell cycle, cancer stem cell, microenvironment, microinvasion, DNA integrity

## Abstract

**Simple Summary:**

The cell cycle regulator Cyclin E1 is a key mediator and biomarker of liver cancer progression in mice and man independent of its canonical interacting partner Cyclin-dependent kinase 2. Over-expression of Cyclin E1 during hepatocarcinogenesis modulates several distinct biological processes such as proliferation, DNA damage response, stemness, invasion and the tumour microenvironment. Interventional depletion of Cyclin E1 in the course of liver cancer progression significantly reduces tumour burden. In contrast, the expression of Cyclin-dependent kinase 2 is dispensable for the progression of liver cancer in mice and lacked diagnostic or prognostic value in patients. Thus, specific inhibition of Cyclin E1 expression represents a promising strategy for the treatment of liver cancer.

**Abstract:**

Cyclin E1 (CCNE1) is a regulatory subunit of Cyclin-dependent kinase 2 (CDK2) and is thought to control the transition of quiescent cells into cell cycle progression. Recently, we identified CCNE1 and CDK2 as key factors for the initiation of hepatocellular carcinoma (HCC). In the present study, we dissected the contributions of CCNE1 and CDK2 for HCC progression in mice and patients. Therefore, we generated genetically modified mice allowing inducible deletion of *Ccne1* or *Cdk2*. After initiation of HCC, using the hepatocarcinogen diethylnitrosamine (DEN), we deleted *Ccne1* or *Cdk2* and subsequently analysed HCC progression. The relevance of CCNE1 or CDK2 for human HCC progression was investigated by in silico database analysis. Interventional deletion of *Ccne1*, but not of *Cdk2*, substantially reduced the HCC burden in mice. *Ccne1*-deficient HCCs were characterised by attenuated proliferation, impaired DNA damage response and downregulation of markers for stemness and microinvasion. Additionally, the tumour microenvironment of *Ccne1*-deficient mice showed a reduction in immune mediators, myeloid cells and cancer-associated fibroblasts. In sharp contrast, *Cdk2* was dispensable for HCC progression in mice. In agreement with our mouse data, *CCNE1* was overexpressed in HCC patients independent of risk factors, and associated with reduced disease-free survival, a common signature for enhanced chromosomal instability, proliferation, dedifferentiation and invasion. However, *CDK2* lacked diagnostic or prognostic value in HCC patients. In summary, CCNE1 drives HCC progression in a CDK2-independent manner in mice and man. Therefore, interventional inactivation of CCNE1 represents a promising strategy the treatment of liver cancer.

## 1. Introduction

Hepatocellular carcinoma (HCC) is the fourth most common cause of cancer-related death and classically develops in a multi-step process based on chronic hepatitis, caused, e.g., by viral hepatitis, alcohol consumption or non-alcoholic metabolic injury [1]. During these processes, hepatocytes accumulate oncogenic mutations due to genetic instability or viral insertions in cell cycle-related pathways, such as *MYC*, *TP53*, Retinoblastoma protein (*RB*) and cyclins [2]. Genomic instability and mutations trigger the hallmarks of cancer during HCC progression (e.g., proliferation, metastasis, immune evasion), but also shape drug resistance and therapeutic efficacy, mediated by aberrant DNA damage repair capacity or activation of cancer stem cells [3,4]. Surgical (liver resection, transplantation) or locoregional (e.g., transarterial chemoembolisation) HCC treatments are only curative for patients in early stages of the disease. In the last decade, improved drug therapy agents, such as tyrosine kinase inhibitors (e.g., Regorafenib), selective Cyclin-dependent kinase (CDK) inhibitors (e.g., Palbociclib) or immunotherapeutic strategies (e.g., Atezolizumab, Nivolumab) have shown partial benefits for patients with advanced HCC, but the overall prognosis remains limited [5]. For instance, in patients with progressive HCC after Sorafenib-treatment, Regorafenib has been reported to provide an overall survival benefit of only 10.6 months versus 7.8 months compared with placebo. Recent data on Regorafenib or Nivolumab indicated that pharmacokinetics, toxicity and clinical responses highly depend on gene polymorphisms (reviewed in [5]).

Cyclins are oscillatory expressed during different phases of the cell cycle to modulate catalytically active CDKs in specific complexes. E-type cyclins (CCNE1, CCNE2), together with Cyclin-dependent kinase 2 (CDK2), were described to regulate transition from the Gap1 (G1)-phase of the cell cycle into DNA-synthesis (S-phase). Additionally, CDK2 together with Cyclin A triggers S-phase progression [6]. Recently, we demonstrated that *Ccne1* is a suitable therapeutic target for the treatment of liver fibrosis [7]. In addition, *Ccne1* and *Cdk2*, but not *Ccne2*, are essential for HCC initiation in vivo. We and others provided evidence for substantial roles of E-type Cyclins during HCC progression, which are associated with impaired overall patient survival and a specific genetic signature characteristic of high replication stress and strong DNA repair response—potentially promoting genetic instability [2,8,9,10]. Several studies indicated that deregulated gene expression, structural variants and aberrant degradation signalling shape the oncogenic functions of CCNE1, which appear to be at least partially CDK2-independent [9,10,11]. In fact, the individual contributions of CCNE1 and CDK2 for the progression of HCC in vivo and their relevance as diagnostic and prognostic biomarkers or therapeutic targets in patients have not been addressed in detail so far.

In this study we aimed to dissect the precise contribution of CCNE1 and CDK2 specifically for HCC progression. As a basis for future therapeutic approaches, we aimed at evaluating whether its interventional deletion after onset of HCC might be beneficial. We will show that targeting of *Ccne1* alone affects several important hallmarks of HCC progression and significantly reduces tumour burden at an advanced stage. Surprisingly, we could also obtain clear evidence that the oncogenic functions of CCNE1 during HCC progression in mice and man are largely CDK2-independent. Finally, we validated that the expression of *CCNE1*, but not *CDK2*, may serve as a diagnostic and prognostic biomarker in HCC patients.

## 2. Materials and Methods

### 2.1. Animal Experiments

This study used transgenic C57/BL6 mice with floxed alleles of *Ccne1* or *Cdk2* [12,13]. To generate conditional *Ccne1* or *Cdk2* knockout animals, mice were crossed with transgenic mice expressing *Cre*-recombinase under control of the *Mx1* promoter [14] as described earlier [9]. Animals were housed in a temperature-controlled room with 12 h light/dark cycles and free access to food and water. Experiments were carried out according to the German legal requirements and were approved by the authority for environment conservation and consumer protection of the state North Rhine-Westphalia (LANUV, Germany). For HCC induction, fourteen-day-old male mice were once injected intraperitoneally (i.p.) with 25 mg diethylnitrosamine (DEN, N0756, Sigma-Aldrich, St. Louis, MO, USA)/kg body weight. DEN-treated mice received three i.p. injections of 400 µg poly-I:poly-C (pIpC, P1530, Sigma-Aldrich, St. Louis, MO, USA) for 5 consecutive days at the age of 24 weeks, inducing the deletion of *Ccne1* or *Cdk2* exclusively in *Mx*-*Cre*-positive littermates. Mice were analysed after week 40.

### 2.2. Tumour Quantification

Livers were removed, separated into individual lobes and analysed for the number and diameter of visible HCC nodules.

### 2.3. Quantitative Real-Time PCR (qPCR)

Total RNA from macroscopically dissected superficial tumorous tissues of lesions with a minimal size of approximately 5 mm and remnant surrounding liver tissues with smaller lesions (smaller than 5 mm) was separately isolated using peqGOLD RNAPure (30-1010, Peqlab, Erlangen, Germany), quantified at 260 and 280 nm with Nanodrop Lite (ND-LITE-PR, Thermo Fisher Scientific, Waltham, MA, USA) and reverse transcription into cDNA by Omniscript RT Kit (205113, Qiagen, Hilden, Germany) was performed as described recently [7]. Relative quantitative gene expression was measured via real-time PCR using a QuantStudio 6 Flex Real-Time PCR System with Real-Time PCR Software v1.3 (Applied Biosystems, Foster City, CA, USA). Target gene expression was normalised to Glyceraldehyde 3-phosphate dehydrogenase (*Gapdh*) expression as internal standard and calculated as fold induction in comparison to controls as indicated. All primer sequences used for qPCR (Eurofins Genomics, Louisville, KY, USA) are given in Appendix A.

### 2.4. Histology and Immunoblot (IB) Analysis

Immunohistochemistry, hematoxylin and eosin (H&E)-staining, isolation of whole cell extracts from macroscopically dissected superficial tumorous tissues of lesions with a minimal size of approximately 5 mm and remnant surrounding liver tissues with smaller lesions (smaller than 5 mm) or human samples, and Western blot analysis were performed as described previously [10,15]. Antibodies used for probing are listed in Appendix A. Primary antibodies for Western blot were used at a range of 1:1000–1:500. Immunohistochemistry was performed with the primary antibodies against Ki67 (ab16667, Abcam, Cambridge, UK, 1:200) or ACTA2 (A2547, Sigma-Aldrich, St. Louis, MO, USA, 1:250) and immunofluorescence staining was performed with a primary antibody against pH2ax (#9718, Cell signalling, Danvers, MA, USA, 1:250) or Ki67 (ab16667, Abcam, Cambridge, United Kingdom, 1:200) and counterstaining by DAPI (H-1200-10, Vectashield mounting medium with DAPI, Vector Laboratories, Burlingame, CA, USA). As secondary antibodies, mouse anti-rabbit IgG-HRP (sc-2375, Santa Cruz Biotechnology, Dallas, TX, USA, 1:5000), m-IgGλ BP-HRP (sc-516132, Santa Cruz Biotechnology, Dallas, TX, USA, 1:5000), m-IgGκ BP-HRP (Santa Cruz Biotechnology, Dallas, TX, USA, 1:5000), ImmPRESS HRP Goat Anti-Rabbit IgG Polymer Detection Kit (MP-7451-15, Peroxidase, Vector Laboratories, Burlingame, CA, USA, according to manufacturer’s recommendation), Alexa Fluor 488 goat anti-rabbit (A-11008), Alexa Fluor 594 donkey anti-rabbit (A-21207, Invitrogen Thermo Fisher Scientific, Waltham, MA, USA, 1:400–500) were used (see Appendix A). For immunoblot analysis, determination of GAPDH (MCA4739, AbD Sero-Tec, Düsseldorf, Germany) or beta actin (ab8227, Abcam, Cambridge, United Kingdom) expression was used as loading control. Discrimination between human non-tumorous liver, matching HCC and adenoma in small human sample cohort was validated by heat-shock protein 70 (HSP70, MAB1663, R&D Systems, Minneapolis, MN, USA) expression, an accepted biomarker for early HCC diagnosis [16]. Immunoblots were visualised by ECL Prime Western Blotting System RPN2232 using an ImageQuant LAS 4000 System with LAS 4000 v1.2 Control Software (GE Healthcare, Chicago, IL, USA), according to the manufacturer’s protocol (see Appendix A). Stained microscopic images were acquired at 200× magnification with a Zeiss Axio Imager.Z1 microscope and AxioVision SE64 Rel. 4.9.1 software (Carl Zeiss, Jena, Germany). Per liver tissue 8–10 images/HPFs (HPF = 196,279 µm^2^) per mouse and 3–10 images/HPFs per individual intrahepatic or explanted tumour were randomly analysed.

### 2.5. Immunoprecipitations (IP), In Vitro Kinase Assays and Enzyme Linked Immunosorbent Assay (ELISA)

Immunoprecipitations and kinase assays were performed as described previously [15] with minor modifications. Briefly, 250–500 μg of protein extract was incubated with 2 µg of antibodies against CDK2 (sc-6248, Santa Cruz Biotechnology, Dallas, TX, USA) or CCNE1 (sc-247 or sc-481, Santa Cruz Biotechnology, Dallas, TX, USA) overnight at 4 °C in Nonidet P-40 lysis buffer containing protease inhibitors (11836170001, cOmpleteTM Mini, Roche, Mannheim, Germany) and phosphatase inhibitors (04906837001, PhosSTOP EASYpack, Roche, Mannheim, Germany). Antibody/protein complexes were coupled to protein A/G agarose beads (sc-2003, Santa Cruz Biotechnology, Dallas, TX, USA) for 1 h at 4 °C and subsequently washed three times in NP40-lysis buffer and in kinase buffer (50 mM HEPES pH 7.5; 10 mM MgCl_2_; 10 mM ß-Glycerophosphate; 1 mM DTT), respectively. Kinase assays were performed by incubating the immunoprecipitated protein complexes in kinase buffer containing 15 μM ATP, 160 Bq [γ-32P]ATP (SRP-401, Hartmann, Braunschweig, Germany) and 2 μg of recombinant histone H1 (10223549001, Roche, Mannheim, Germany) in a total volume of 30 µL for 30 min at 4 °C. The reaction was inactivated with SDS/PAGE sample buffer and resolved by electrophoresis on polyacrylamide gels. Immobilised gels were exposed to autoradiography (28-9068-44, Hyperfilm, GE Healtcare, Freiburg, Germany); see Appendix A. ELISA from liver protein extracts (1 µg/µL) in NP40-lysis buffer containing protease and phosphatase inhibitors (see above) for CCL2 (88-7391-86, Mouse CCL2 ELISA Ready-SET-Go!, eBioscience, San Diego, CA, USA), IL6 (88-7064-86, Mouse IL-6 ELISA Ready-SET-Go!, eBioscience, San Diego, CA, USA), INFG (88-7314-88, Mouse INF gamma ELISA Ready-SET-Go!, eBioscience, San Diego, CA, USA) and IL10 (88-7105-86, Mouse IL-10 ELISA Ready-SET-Go!, 2nd Generation, eBioscience, San Diego, CA, USA) were performed according to the manufacturer’s recommendations and analysed with a Cytation 3 Cell Imaging Reader using Gen5 v3.00.19 Software (Biotek Instruments, Winooski, VT, USA).

### 2.6. Fluorescence-Activated Cell Sorting (FACS)

FACS-based analysis of the immune cell compartment in liver lobes displaying at least one dysplastic lesion was performed as described recently [17]. Co-staining of cells was performed using a mix of fluorescence-labelled antibodies diluted in FACS-blocking buffer (mixture of 0.66% human/rabbit/mouse serum, Sigma-Aldrich, St. Louis, MO, USA, and 1% Bovine Serum Albumin, Sigma-Aldrich, St. Louis, MO, USA, in 1× PBS) for 30 min at 4 °C. Antibody details are listed in Appendix A. Compensation of each fluorochrome was automatically performed using OneComp eBeads (01-1111-42, Invitrogen Thermo Fisher Scientific) according to manufacturer’s recommendations. Measurements were performed using a BD LSRFortessa (BD Bioscience, San Jose, CA, USA) and data were analysed using FlowJo 7.6.5 Software (Tree Star, Ashland, OR, USA). A gating strategy was carried out as recently demonstrated with minor modifications [17,18]. During myeloid subgating, liver-resident macrophages (Kupffer cells, Hoechst^−^CD45^+^CD11b^INT^F4/80^+^) were excluded before gating on infiltrating MDSC and TAM subpopulations (Appendix A).

### 2.7. Processing of Human and Murine Liver/HCC Samples

Human liver samples from 5 patients were investigated for CDK2 kinase activity. The anonymous samples were obtained from the Institute of Pathology of the University Hospital RWTH Aachen after surgical liver/HCC resection. The cohort consisted of 3 patients with moderately differentiated (G2) HCC (matching tumour and liver tissue) and 2 patients with liver adenoma. Healthy control liver tissue was obtained from a donor liver not used for transplantation and from biopsies of healthy subjects with histologically normal livers. The study protocol was approved by the ethics committee of the University Hospital RWTH Aachen and conducted according to the principles expressed in the Declaration of Helsinki. A more detailed overview on patient characteristics is given in Appendix A.

Liver and tumour samples from DEN-treated mice were processed as follows: Livers were investigated for tumour numbers and size of nodules. Tissue was then dissected as follows (compare also Figure 1a): tumorous tissues of lesions with a minimal size of approximately 5 mm, remnant surrounding liver tissue with smaller lesions (smaller than 5 mm) and pieces of liver lobes of approximately 0.2–0.4 g displaying a minimum of one lesion were dissected. Dissected tumorous and liver tissues were preserved by embedding in paraffin for detailed investigation of tumour-free liver tissue versus individual intrahepatic and explanted tumorous tissues by IHC in cooperation with an experienced pathologist (N.G., analysis of mitotic cells and vascular invasion V1, Appendix A). Further preservation of dissected tumorous and liver tissues by cryoconservation allowed investigation of whole liver tissue (including small lesions) and separated tumour tissue by qPCR, immunoblot, immunofluorescence and ELISA assay. Of note, this mode of preservation limits the level of investigation, as exclusion of intrahepatic HCC tissues is limited compared to paraffin-embedded tissues. Freshly isolated liver lobes displaying a minimum of one lesion were subjected to enzymatic digestion and FACS-based analysis of living CD45 positive myeloid subpopulations. Livers from genetically unmodified and untreated C57/BL6 wildtype mice at the age of eight weeks (*n* = 3) were included in the qPCR analysis as an indicator for an efficient separation of surrounding liver and tumorous tissue in the DEN model (Ctrl, excluded from statistical analysis).

Generation of HCC samples from Dimethylbenz[a]anthracene (DMBA) treated and Western diet (WD)-fed mice and from ASV-B transgenic mice were performed by the Department of General, Visceral and Transplantation Surgery of the University Hospital RWTH Aachen. Male C57BL/6 mice received 50 µL 0.5% DMBA (7,12-Dimethylbenz[a]anthracene) dissolved in acetone to the dorsal surface on the fourth postnatal day and were fed sucrose-enriched water (10%) and a WD with 40 kcal% fat, 20 kcal% fructose and 2 kcal% cholesterol (D09100301, Research Diets, New Brunswick, NJ, USA) for 30 weeks and tumour lesions were isolated [19]. ASV-B transgenic mice, expressing the simian virus 40 large and small T antigens under the control of the antithrombin III regulatory sequences [20], were sacrificed at the age of 16 weeks, followed by the isolation of tumour nodules.

### 2.8. Measurement of Serum Enzyme Activity

Alanine aminotransferase (ALT) and aspartate aminotransferase (AST) activities were measured in murine serum by standard methods (UV test at 37 °C, central diagnostic laboratory Aachen University Hospital) using an automated Roche Modular preanalytics system (Hoffmann-La Roche).

### 2.9. Validation of Ccne1 and Cdk2 Deletion

Genotyping of mice and determination of knockout after pIpC treatment was confirmed on genomic level by PCR. Liver samples were incubated in lysis buffer at 65 °C overnight and genomic DNA was isolated and PCRs were performed as described previously [21]. All PCR primer sequences, primer combinations and wildtype (+/+), loxP-flanked (f/f) and knockout (−/−) amplification products are given in Appendix A.

### 2.10. Analysis of Human TCGA, TCPA and GTex Databases

Bioinformatic analysis of CCNE1 and CDK2 expression in HCC patients was based on RNA-seq data and reverse phase protein array (RPPA) generated by the The Cancer Genome Atlas (TCGA) Research Network, LIHC cohort (https://www.cancer.gov/tcga/, 373 HCC samples and 50 matching non-tumorous samples, accessed on 5 May 2021), The Cancer Proteome Atlas (TCPA), LIHC cohort (https://www.tcpaportal.org/, 184 HCC samples analysed with 218 protein structures via antibodies, accessed on 5 May 2021) and Genotype-Tissue Expression (GTEx) project (http://gtexportal.org/, liver cohort, 110 non-tumorous patient samples, accessed on 5 May 2021) [22,23,24]. Extended data, analyses and visualisation were based on clinical, mRNA and protein expression data available at cBioPortal (http://cbioportal.org/, mRNA expression and aneuploidy scores for correlation analyses, RPPA data of CcnE1 in HCC, accessed on 5 May 2021), GEPIA2 (http://gepia.cancer.org/, mRNA expression between liver and tumour, mRNA expression in iClusters, disease-free survival, accessed on 5 May 2021) or TCPA (https://www.tcpaportal.org/, Visualisation of RPPA data in HCC, RPPA corelation analysis in HCC, accessed on 5 May 2021) [23,24,25,26,27]. HCC patient aetiologies/risk factors were classified as previously described [28]. DFS defines the length of time after primary treatment for a cancer ends that the patient survives without any signs or symptoms of that cancer (National Cancer Institute Dictionary of Cancer Terms, U.S. Department of Health and Human Services, National Institutes of Health, available at https://www.cancer.gov/publications/dictionaries/cancer-terms/, accessed on 5 May 2021). The DFS curves are based on the median expression (cutoff-high/low: 50%) of each relevant gene (*CCNE1* and *CDK2*), dividing the patients into low- and high-expressing groups according to GEPIA2. Data were analysed as batch normalised from Illumina HiSeq_RNASeqV2 RSEM (log2) values or TPM + 1 (log2) values according to cBioPortal or GEPIA2. Patients were further analysed by mRNA expression and histological tumour grade (G1, *n* = 55; G2, *n* = 1 77; G3, *n* = 119; G4, *n* = 12). Normalisation of the gene-based count data was performed using the variance stabilisation method from the *DESeq2* R package [29]. *p*-values were obtained by applying the Wilcoxon rank sum test to compare pairs of grades. Patients with grade 4 were not investigated further due to the low number of available samples. Indicated *p*-values have been adjusted using the Benjamini–Hochberg multiple test correction.

### 2.11. Availability of Whole Transcriptome Shotgun Sequencing Data (RNA-Seq) from DEN-Derived Hepatoma Cell Lines

The generation of hepatoma cell lines from solid tumour tissue (HCC cells) and precancerous liver tissue (preCL cells) of 40-week-old mice, DEN-treated and subsequent analysis of data was previously described [10]. The data reported in this paper have been deposited in the Gene Expression Omnibus (GEO) database, https://www.ncbi.nlm.nih.gov/geo/ (accession no. GSE111079, accessed on 22 October 2021).

### 2.12. Statistical Analysis

Wherever applicable, data were depicted in scatter plots with bars. Accordingly, each dot indicates the data from one individual animal or tissue sample. Statistical outliers were identified once using Graph Pad Prism 7 ROUT-method (GraphPad Software, San Diego, CA, USA). Statistical significance was determined using Graph Pad Prism 7 (GraphPad Software, San Diego, CA, USA). The normal distribution was analysed by D’Agostino–Pearson test before each parametric or non-parametric test. Analysis between two groups was performed by an unpaired, two-tailed parametric t-test or an unpaired, two-tailed non-parametric Mann–Whitney test. Comparisons between multiple groups was performed via unpaired one-way analysis of variance (ANOVA) with Tukey’s multiple comparison test (parametric) or unpaired Kruskal–Wallis test with Dunn´s method (non-parametric). Comparison of vascular invasion was performed by Chi-square test and Fisher’s exact test. Disease-free survival analysis was performed with the Mantel–Cox test (according to http://gepia2.cancer-pku.cn/, accessed on 5 May 2021). A Wilcoxon rank sum test was performed to compare pairs of tumour grades. All data are presented as mean ± SD. The correlations were analysed using Pearson (parametric) or Spearman (non-parametric) correlation. Significances were defined as *: *p* < 0.05; **: *p* < 0.01; ***: *p* < 0.001.

## 3. Results

### 3.1. Interventional Deletion of Ccne1, but Not of Cdk2, after Onset of HCC Inhibits Tumour Progression

To assess the specific contributions of CCNE1 or CDK2 for HCC progression, DEN was injected into two-week-old conditional *Ccne1* or *Cdk2* knockout mice with inducible *Cre*-recombinase under control of the *Mx1*-gene promoter [14]. It was previously shown that the molecular signature of DEN-induced tumours reflects the situation of human HCC subclasses with poor prognosis [30]. Further, earlier results demonstrated that DEN-treatment per se does not induce *Mx-Cre*-mediated deletion of floxed target genes [31]. After HCC initiation (22 weeks), interventional inactivation of *Ccne1* (Ccne1^−/−^) or *Cdk2* (Cdk2^−/−^) in liver, lymphocytes and hematopoietic cells was induced by pIpC injection. *Cre*-negative littermates were treated equally and served as wildtype (WT) controls (Ccne1^f/f^, Cdk2^f/f^, Figure 1a). In consensus with previous reports [9], pIpC treatment efficiently deleted *Ccne1* or *Cdk2* in 22-week-old DEN-treated livers (Appendix A). Sixteen weeks after intervention (week 40), mice were sacrificed and subjected to extensive analyses.

Relative liver weights, as well as activities of alanine transaminase and aspartate transaminase in serum, were not altered by targeted deletion of *Ccne1* or *Cdk2* (Appendix A). Macroscopically and histologically, all mice developed HCCs (Figure 1b). However, interventional deletion of *Ccne1* resulted in significantly reduced tumour numbers and the cumulative tumour size of approximately 50% compared to controls (Figure 1c), but had no impact on the mean or median tumour size (Appendix A). In contrast, interventional deletion of *Cdk2* had no effect on tumour burden compared to Cdk2^f/f^ mice (Figure 1c and Appendix A). Altogether these data indicate that CCNE1, but not CDK2, is a key factor in progression from early stage to advanced HCCs.

In order to characterise the critical role of CCNE1 for HCC growth, we investigated the cell cycle activity and regulation in advanced HCCs 16 weeks after intervention. The cell cycle marker protein Ki67 displayed heterogeneous intertumoral and intratumoral expression in DEN-induced WT HCCs (Appendix A). Although, Ki67 was highly expressed in DEN-induced WT HCCs compared to the corresponding non-tumorous tissues (Figure 2a,b). Importantly, the deletion of *Ccne1*, but not of *Cdk2*, significantly reduced the number of Ki67-positive cells in tumours (Figure 2a,b). We performed detailed gene expression analysis of cyclins controlling G1-phase (Cyclin D1, *Ccnd1*), G1/S-phase (*Ccne1*, *Ccne2*) and S-phase (Cyclin A2, *Ccna2*). *Ccne1*, *Ccnd1* and *Ccna2* were significantly induced in tumours compared to surrounding liver tissues of Ccne1^f/f^ and Cdk2^f/f^ animals (Figure 2c). As expected, 16 weeks after inducing *Cre*, *Ccne1* mRNA levels were strongly reduced in tumours of Ccne1^−/−^ mice compared to controls. Interestingly, *Ccne1*-depleted tumours also displayed reduced levels of *Ccnd1* and *Ccna2* compared to WT mice (Figure 2c). Importantly, targeted deletion of *Cdk2* did not result in reduction of any of these cyclins (Figure 2c). *Ccne2* expression was not significantly regulated in tumours and livers of all cohorts (Figure 2c). Liver tumours of Ccne1^f/f^ mice revealed enhanced mitosis compared to surrounding tumour-free liver tissues. In contrast, the number of mitotic cells were not elevated in Ccne1^−/−^ tumours (Figure 2d, Appendix A). Altogether, targeted deletion of *Ccne1*, but not of *Cdk2*, reduced cell cycle activity in advanced HCCs. These data indicate that CCNE1 is a central mediator of HCC progression and proliferation, which cannot be compensated by other cyclins.

### 3.2. Absence of Ccne1 Modifies the DNA Damage Response and Restrains Stemness and Microinvasion during HCC Progression

Aberrant hepatic CCNE1 expression induces DNA damage due to replication stress and oxidative stress, which promotes double-strand breaks (DSB), chromosomal instability and selective hepatocyte transformation in mice [2,8,10]. Hence, we investigated the influence of interventional *Ccne1* or *Cdk2* depletion in HCC on markers of DNA damage and repair during HCC progression. Phosphorylation of histone H2AX (pH2AX), a marker for DSB, in dissected tumours and corresponding liver tissues was not significantly different between Ccne1^f/f^ and Ccne1^−/−^ mice at week 40 after DEN (Figure 3a,b).

We next examined the extent of general DNA damage responses (DDR). DSB or DNA single-strand breaks (SSB) activate the Ataxia telangiectasia mutated/Ataxia telangiectasia and Rad3 related (ATM/ATR) pathway. In HCC, overexpression of *CCNE1* is associated with ATR activation and upregulated DNA repair genes [2,8]. Deletion of *Ccne1*, but not Cdk2, in advanced liver tumours was associated with reduced phosphorylation of an unknown ATM/ATR substrate of approximately 35 kDa (Figure 3c). In DEN-induced WT HCC, the expression of the DNA mismatch repair gene MutS Homolog 2 (*Msh2*) and the DSB repair gene *Rad51b* were significantly induced compared to surrounding liver tissues. Interventional inactivation of *Ccne1* significantly reduced *Msh2*, *Rad51b* and *Rad21* expression in tumour tissues (Figure 3d). In addition, inactivation of *Ccne1* led to a downregulation of *Msh2* in the liver already under basal conditions (Figure 3d). Interestingly, interventional *Cdk2* deletion triggered a slightly different DDR, as it resulted in downregulation (not significant) of *Rad51b* and *Rad21* in tumours, but not in deregulation of *Msh2* (Figure 3e). These data illustrate that DDR during HCC progression is attenuated after depletion of *Ccne1*, and to a lesser extent by inhibiting *Cdk2*.

Subsequently, we investigated the relationship between *Ccne1* expression and the stemness-associated HCC markers alpha fetoprotein (*Afp*) and prominin 1 (*Prom1*; CD133) [32]. DEN-treated WT mice revealed a high induction of *Afp* and *Prom1* in HCCs compared to corresponding livers. Inactivation of *Ccne1* during early HCC progression prevented upregulation of these stemness markers in liver tumours (Figure 4a). In contrast, induction of *Afp* or *Prom1* expression in HCCs was still observed after interventional inactivation of *Cdk2* (Figure 4b).

We further investigated a potential role of *Ccne1* inactivation on invasive properties of HCC. Approximately 60% of the investigated DEN-treated WT livers, but none of the Ccne1^−/−^ animals displayed hepatic microvascular invasion (V1) [33] of hepatoma cells after histological investigation (Figure 4c, Appendix A). In good agreement, we found induction of the cell motility and invasion-associated marker genes Rb Binding Protein 7 (*Rbbp7*, Appendix A), Intracellular adhesion molecule 1 (*Icam1*) and Epithelial cell-transforming sequence 2 (*Ect2*) [34,35,36] in WT HCCs compared to the surrounding liver tissue (Figure 4d,e). Importantly, inactivation of *Ccne1* in HCCs substantially diminished this upregulation of invasion-associated genes (Figure 4d,e and Appendix A). Moreover, the expression of the metastasis marker snail family transcriptional repressor 1 (*Snai1*) [37] was found to be downregulated in tumorous and surrounding tissues in absence of *Ccne1* (Figure 4e). Of note, deletion of *Cdk2* showed no impact on expression levels of *Ect2* or *Snai1* in HCC or surrounding liver tissues (Figure 4f). Taken together, inactivation of *Ccne1* after HCC initiation exhibited a lower reaction to DNA damage and reduced hallmarks of cancer progression, such as stemness and invasion.

Finally, we validated the investigated markers from our in vivo study in already reposited own in vitro-based Whole Transcriptome Shotgun Sequencing data (GEO database, accession no. GSE111079, accessed on 22 October 2021) of early and advanced HCC cells [10]. The previously established and described DEN-derived, murine hepatoma cell lines were ex vivo isolated from solid tumours, referred as HCC cells (HCC, high proliferation rate and *Afp* levels), or surrounding liver tissues, referred as precancerous liver cells (preCL, low proliferation rate and *Afp* levels) of 40-week-old, DEN-treated animals and compared to non-transformed primary isolated hepatocytes ([10] and Figure 4g). For the current study, we reinvestigated the data and validated, that markers for proliferation (*Mki67*, *Ccne1*, *Cdk2*, *Ccnd1*, *Ccna2*, *Ccne2*), DNA damage response (*Msh2*, *Rad21*, *Rad51b*), stemness (*Prom1*) and invasion (*Ect2*, *Icam1*, *Rbbp7*, *Snai1*) were specifically associated with DEN-induced hepatocarcinogenesis ([10], Figure 4g). More importantly, the DNA damage marker *Rad51b* and the invasion markers *Icam1*, *Rbbp7* and *Snai1* were significantly induced or at high levels only in advanced HCC cells (Figure 4g). In summary, we demonstrated that our proposed in vivo gene signature for *Ccne1*-associated progression is specific to early or advanced stages of DEN-mediated parenchymal HCC development.

### 3.3. Loss of CCNE1 after HCC Establishment Limits the Accumulation of Activated Myofibroblasts and Myeloid Cells in the Hepatic Microenvironment

Hallmarks of HCC progression are further influenced by interactions with cells of the hepatic stroma, such as extracellular matrix-producing myofibroblasts or immune cells [38]. We therefore investigated whether ameliorated HCC progression in absence of *Ccne1* in liver cells, leucocytes and hematopoietic cells affects the HCC microenvironment. Tumorous tissues from DEN-induced WT mice revealed enhanced numbers of activated, smooth muscle actin alpha 2 (ACTA2)-expressing myofibroblasts compared to non-tumorous liver tissues (Figure 5a,b). Inactivation of *Ccne1* during hepatocarcinogenesis resulted in a significantly reduced number of activated myofibroblasts in tumorous tissues (Figure 5a,b). Consistently, we found that the enhanced expression of the matrix protein collagen type I alpha-1 chain (*Col1a1*) in tumours compared to the surrounding liver tissue was diminished after inactivation of *Ccne1* (Figure 5c). Moreover, deletion of *Ccne1* was associated with a significantly decreased amount of CD45-positive leukocytes compared to livers of Ccne1^f/f^ mice (Figure 5d). Intrahepatic immune cell subpopulation analyses exhibited lowered quantity of myeloid-derived suppressor cells (MDSCs) of monocytic (M-MDSCs) and granulocytic origin (G-MDSCs) in Ccne1^−/−^ mice (Figure 5d and Appendix A). Concomitantly, *Ccne1*-depleted HCCs displayed less tumour-associated macrophages (TAM) positive for MHC-II (TAM1), which are considered as proinflammatory [17] (Figure 5d). Of note, recruited MHC-II-negative TAM (TAM2) or resident Kupffer cell (KC) subpopulations were unchanged by loss of *Ccne1* (Appendix A). Deletion of *Cdk2* during HCC progression had no effect on the number of CD45-positive cells or myeloid subpopulations (Appendix A). Altogether, absence of *Ccne1* is associated with a reduction in myofibroblasts and infiltrating myeloid cells during HCC progression.

In consensus with these findings, CD45-positive leukocytes and TAM1 negatively correlated with tumour size in the *Ccne1*-depleted cohort, which was not the case in control or Cdk2^−/−^ animals (Figure 5e and Appendix A). This finding suggests that CCNE1 might be involved in controlling immune cell infiltration to the HCC environment. Subsequently, we examined potential mediators of MDSC and TAM1 recruitment, such as CC chemokine ligands 2 and 5 (*Ccl2*, *Ccl5*) and the CC chemokine receptors 2 and 5 (*Ccr2*, *Ccr5*) [17]. Expression levels of *Ccl2* and *Ccl5* were not significantly altered between tumours and corresponding liver tissues from Ccne1^f/f^ and Ccne1^−/−^ animals (Appendix A). In addition, the hepatic CCL2 and interferon gamma (INFG) protein levels were unchanged after the depletion of *Ccne1* or *Cdk2* in DEN-treated mice, which excludes an altered potential for chemotaxis or T cell-mediated activation of myeloid cells (Appendix A). Importantly, mRNA levels of *Ccr2* and *Ccr5*, known to be differentially expressed by TAM1 and MDSCs [17], were enhanced in isolated control tumours compared to liver tissues, postulating an enhanced presence of *Ccl2* and *Ccl5*-responding myeloid cells (Figure 5f). Ccne1^−/−^ tumours revealed diminished levels of *Ccr2* and *Ccr5* (Figure 5f), which further indicates the decrease in infiltrating myeloid subpopulations in absence of CCNE1.

*Interleukin (Il) 10, Il6* or *Cd274* (PD-L1) can basically be expressed by hepatoma cells, cancer-associated fibroblasts, TAMs or MDSCs, which promote tumour growth, metastasis and immune evasion [38,39]. However, in WT mice we did not detect upregulation of *Il10* or *Cd274* in DEN-induced HCCs (Figure 5g). Interestingly, interventional deletion of *Ccne1* during hepatocarcinogenesis was associated with significantly reduced *Il10* and *Cd274* mRNA levels in non-tumorous liver samples (Figure 5g). Accordingly, we found diminished hepatic protein levels of IL10 and IL6 in DEN-treated Ccne1^−/−^, but not in Cdk2^−/−^ animals at week 40 (Figure 5h and Appendix A). This suggests that *Ccne1* inactivation triggers decreased immunoregulatory effects, such as immunosuppression, during advanced hepatocarcinogenesis.

The data showed so far that HCC progression in the DEN model requires an induction of *Ccne1*, which in turn is associated with upregulation of DDR genes, stemness factors, markers of metastasis and myeloid infiltration. We examined these relationships in two additional mouse HCC models comprising either application of the carcinogen DMBA together with a Western diet (DMBA/WD) [19] or ASV-B transgenic mice [20] as illustrated in Appendix A. All investigated tumour tissues displayed enhanced *Ccne1* gene expression (Appendix A). Importantly, both models displayed induction of at least one DNA repair gene (*Rad21*, *Rad51b*, or *Msh2*, Appendix A), while the stem cell markers *Afp* and *Prom1* were highly expressed only in the DMBA/WD model (Appendix A). However, the invasion-associated marker *Ect2* and the chemokine receptor *Ccr5* were induced in both HCC models (Appendix A). These data suggest that *Ccne1* induction is a general model-independent property of murine hepatocarcinogenesis, which is frequently associated with a signature of upregulated markers for DDR, stemness, invasion and leucocyte recruitment.

### 3.4. CCNE1 Expression Is a Diagnostic and Prognostic Biomarker, Correlating with a Molecular Signature of Proliferation, Chromosomal Instability, Dedifferentiation, Invasion and Leucocyte Infiltration in HCC Patients

Using a comprehensive mRNA expression dataset generated by the TCGA Research Network (https://www.cancer.gov/tcga/, accessed on 5 May 2021), we recently demonstrated that high levels of *CCNE1* correlated with reduced overall survival of HCC patients (liver hepatocellular carcinoma, LIHC cohort) after diagnosis, while *CDK2* expression level had no effect on general patient mortality [10]. Here, we extensively expanded the analysis of *CCNE1* and *CDK2* as prognostic and diagnostic markers and therapeutic targets by including data of matching (TCGA, LIHC cohort) and independent (GTEx, liver cohort, http://gtexportal.org/) tumour-free liver samples along with web-based analysis and visualisation tools (accessed on 5 May 2021) [22,25,26,27].

Importantly, *CCNE1* but not *CDK2* expression was significantly enhanced in HCC samples from patients compared to non-tumorous liver specimens (Figure 6a), independent of HCC aetiology (Appendix A). The subgroup of patients displaying high *CCNE1* expression after primary cancer treatment was strongly (*p* = 0.000021) associated with reduced disease-free survival (DFS), pointing to enhanced risk for postcurative mortality in these patients (Figure 6b). In comparison, high expression of *CDK2* only slightly (*p* = 0.049) reduced DFS (Figure 6b). We investigated a potential relationship between the histological HCC grade (serving as a measure for differentiation status and malignancy of tumours) and the level of *CCNE1* and *CDK2* expression. Importantly, *CCNE1* expression was lowest in G1 (well differentiated), moderate in G2 (intermediate grade) and highest in G3 (poorly differentiated) HCCs, suggesting that *CCNE1* level correlates with malignancy (Appendix A). In contrast, *CDK2* expression was similar in G1 and G2 HCCs but elevated in G3 tumours, pointing towards a role during late-stage hepatocarcinogenesis (Appendix A).

We then classified the *CCNE1* and *CDK2* expression in these patients according to demographic, pathological and molecular characteristics of three well-described, different HCC subtypes (integrated clusters iClust1, 2, 3) identified by integrative in silico multi-platform subtyping approaches [40] (see Appendix A and [41]). Compared to tumour-free liver samples, *CCNE1* expression in tumours was significantly upregulated in iClust1 and 3 (Appendix A). Both clusters share overexpression of proliferation marker genes (e.g., Ki67, excessive in iClust1), high tumour grade, high presence of vascular invasion, poor differentiation (highest AFP level in iClust1) according to the Hoshida classification and immune infiltration, in addition to unique histological or oncogenic properties [40,41]. iClust3 shows the highest frequency of chromosomal instability (chromosome arm alterations) and *TP53* mutation. In contrast, iClust2 characterises non-proliferative, low-grade tumours, a low rate of vascular invasion and immune exclusion [40,41]. Importantly, *CDK2* expression was not associated with a defined HCC cluster (Appendix A).

Next, we validated individual human marker gene expression corresponding to our murine data. In good agreement, the HCC-related stemness markers *AFP* and *THY1* (CD90) highly correlated with the expression of *CCNE1* in patients, while the *AFP* correlation with *CDK2* expression was moderate at best (Figure 6c and Appendix A). *CCNE1*, but not *CDK2* expression also correlated with the levels of the invasive markers *ICAM1* and vimentin (*VIM*) [37] and stroma-associated factors, such as leucocyte receptor *CCR5* and the matrix protein *COL1A1* in patients (Figure 6d and Appendix A). Additionally, *CCNE1*, but not *CDK2* expression significantly correlated with the aneuploidy score (Figure 6e), pointing towards a differential contribution of *CCNE1* and *CDK2* expression to chromosomal homeostasis during HCC progression. Summarised, these results clearly demonstrate that *CCNE1*, but not *CDK2* mRNA expression is a diagnostic and prognostic marker in patients referring to an oncogenic signature of proliferation, dedifferentiation, loss of DNA integrity, invasion and stromal signalling.

Finally, we investigated if HCC progression is associated with deregulated protein expression of CCNE1 or CDK2 in patients. The Cancer Proteome Atlas (TCPA, LIHC cohort) database [23,24] revealed that ~46% of the investigated HCC patients (*n* = 82/179) displayed enhanced CCNE1 protein levels above mean (Appendix A). In accordance with our previous findings, CCNE1 protein levels were highly associated with enhanced markers of proliferative signalling, such as AKT, CDK1, ERBB2, MET, NRAS or MYC and levels of markers for DDR (MSH2, RAD51, BRCA2) or metastasis-related signalling (SRC, STAT3, SNAI1) (Appendix A). As CDK2 protein data at large-scale levels are not available so far, we performed functional analyses on CDK2 kinase activity in a small patient cohort of proliferating HCCs with corresponding non-tumorous control tissue of different aetiologies, benign adenomas and healthy control tissue (Appendix A). In consensus with the mRNA expression data (see Figure 6a and Appendix A), CDK2 protein was constitutively expressed in all samples with some variations (Appendix A). All HCC samples displayed distinct CCNE1 protein isoforms (Appendix A). Importantly, subsequent in vitro kinase activity assays revealed that one out of three HCCs did not exhibit CDK2 kinase activity above background levels despite strong tumour proliferation (Figure 6f,g, see sample #243).

Taken together, *CCNE1* expression is enhanced throughout human HCC aetiologies and associated with postcurative mortality and a signature contributing to dedifferentiation, chromosomal instability and invasion, which is independent of *CDK2* levels. We provide evidence that aberrations in *CCNE1* gene regulation and CCNE1 protein expression determine the oncogenic characteristics in humans beyond CDK2 signalling, as proliferation in human HCCs can basically occur without CDK2 kinase activity. These data indicate that CCNE1 is a relevant target in human HCC.

## 4. Discussion

Recent studies have begun to understand the precise role of E-type cyclins and their associated kinase CDK2 for the development of liver cancer. For instance, Geng et al. demonstrated that simultaneous deletion of *Ccne1* and *Ccne2* during early stages of HCC ameliorated disease development [9]. Our own work identified *Ccne1* and *Cdk2,* but not *Ccne2*, as essential for the initiation of HCC [10]. In addition, we and others provided evidence that high *CCNE1* expression in murine and human HCC is associated with DNA damage, a transcriptomic pattern of high DNA repair and poor prognosis [2,8]. However, it was unclear whether inactivation of either CCNE1 or CDK2 at a time when the liver tumour has already progressed might represent a valuable therapeutic strategy.

To address this question, we developed a mouse model that allowed interventional gene targeting of *Ccne1* or *Cdk2* in DEN-treated animals after premalignant dysplastic nodules have been established. Importantly, interventional deletion of *Ccne1* was associated with reduced tumour burden at late stage, while inactivation of *Cdk2* had absolutely no impact on HCC progression at all. This key finding highlights a critical and unique role of CCNE1 for survival and expansion of malignant hepatoma cells in a CDK2-independent manner in vivo, which until now were only hypothesised from in vitro observations [9,10]. Furthermore, the absence of *Ccne1* decreased total cell cycle activity in developed HCCs, illustrating the main regulatory function for modulating CDK2 activity, which cannot be efficiently compensated by other cyclins in mice.

Aberrant or forced hepatic *CCNE1* expression was found to mediate DSB, chromosomal instability and activation of the ATR pathway in response to replication stress and oxidative stress, selectively inducing hepatocyte transformation (initiation) in mice [2,8]. Hence, we tested the relevance of *Ccne1* for DNA damage and DDR specifically during HCC progression. Depletion of *Ccne1* during HCC progression had no effect on the absolute formation of DSB at advanced stage. However, inactivation of *Ccne1* in liver tumours showed decreased phosphorylation of at least one substrate of the ATM/ATR kinases, which indicates that CCNE1 is involved in specific processes of the DDR during HCC progression. The identification of this ATM/ATR substrate was not within the scope of the present study but will be addressed in the future. *Ccne1*-depleted liver tumours further displayed reduced gene expression of markers related to repair of DSB and SSB. Thus, an enhanced DDR response is a hallmark of *Ccne1*-overexpressing HCCs at advanced stage. As *Cdk2*-deficient tumours also displayed low expression of some DNA repair genes, we cannot exclude that distinct functions of CCNE1 for DDR require CDK2.

AFP and PROM1 are accepted markers for cancer stem cells in the liver and are associated with recurrence, metastasis and chemoresistance in HCC [32]. Importantly, these markers were strongly induced in DEN-induced murine liver tumours but diminished after targeting *Ccne1*. In sharp contrast, the inactivation of *Cdk2* had no effect on stemness factor expression. Moreover, deletion of *Ccne1* was associated with reduced expression of *Ect2*, *Rbbp7* [10,34,35], *Icam1* [36,42] and *Snai1* [37], all of which had been described as drivers of metastatic processes and HCC progression. In good agreement, previous studies described that amplification of *CCNE1* was associated with liver metastasis of gastric cancer subtypes [43] and that silencing of *CCNE1* resulted in reduced invasion and migration capabilities in cholangiocarcinoma [44]. Together, these results suggest that CCNE1 is involved in dedifferentiation and acquisition of invasive features of liver cancer cells independent of CDK2. Moreover, we validated our marker signature of DEN-mediated proliferation, DNA damage, stemness and invasion in vitro, using whole transcriptome shotgun sequencing data of previously described [10] murine hepatoma cell lines of early and advanced stage and primary, non-transformed hepatocytes.

HCC progression also involves interactions with the tumour microenvironment [17,38]. In absence of *Ccne1*, HCC progression was characterised by a reduced number of activated myofibroblasts, infiltrating immature MDSCs and differentiated inflammatory TAMs, which was confirmed by reduced mRNA levels of *Col1a1* and the myeloid chemokine receptors *Ccr2*/*Ccr5*. Concomitantly, the immunoregulatory mediators IL6, IL10 and *Cd274*, known to be expressed by HCCs, MDSCs, TAMs and myofibroblasts and to promote protumorigenic signalling on cancer (e.g., growth, angiogenesis, metastasis) or microenvironment (e.g., cell migration, immune evasion) [38,39], were attenuated in *Ccne1*-depleted animals. Two possible explanations should be considered. First, *Mx-Cre*-driven deletion of *Ccne1* in non-parenchymal cells or progenitor cells could directly affect cell infiltration, proliferation or differentiation of immune cells or immune regulatory processes during HCC progression. In line with this hypothesis, we have recently shown that systemic knockdown of *Ccne1* in mice with acute hepatitis resulted in reduced cell cycle activity of myeloid cells [7], and CCNE1 signalling was shown to be involved during myeloid development [45]. In addition, forced overexpression of *Ccne1* in mice resulted in enrichment of several immune system-related functions of inflammatory macrophages in liver (e.g., *Ccr2*, reactive oxygen species, regulation of TNF signalling) [8]. Alternatively, our findings in the HCC environment could be explained by an indirect, secondary effect due to reduced tumour load of *Ccne1*-depleted mice, independent of CCL2-driven chemotaxis or T cell-mediated activation via INFG. However, depletion of *Cdk2* did not show any impact on the microenvironment during HCC progression.

DEN-induced murine HCC in our study were characterised by strong expression of *Ccne1* and induction of markers related to DDR, stemness, invasion and leukocyte infiltration. Importantly, this signature was not restricted to the DEN model, but was also found in two other independent tumour models (DMBA/WD-treated mice and ASV-B transgenic mice). This indicates that the impact of *Ccne1* overexpression on this signature is not model specific but rather represents a general biological principle.

Finally, we aimed to analyse whether our findings regarding CCNE1 and CDK2 in murine liver cancer can also basically be found in HCC patients. In silico analysis of databases with clinical parameters and transcriptome data of HCC patients (TCGA) revealed that *CCNE1*, but not *CDK2*, is highly expressed in human HCCs compared to livers of non-tumorous individuals regardless of risk factors, such as hepatitis virus infections, alcoholic or non-alcoholic liver disease, demonstrating the diagnostic value of *CCNE1* mRNA levels. Our previous study showed that high expression of *CCNE1* was associated with decreased overall patient survival [10]. The present work now identified that high *CCNE1* mRNA levels also reduced the postcurative survival of HCC patients, illustrating *CCNE1* expression as a relevant marker to predict therapeutic prognosis. In line with this, *CCNE1* expression levels also correlated with histological tumour grades, supporting the important function of CCNE1 for differentiation and malignancy during all stages of HCC progression. As *CDK2* expression was aberrant in poorly differentiated G3 tumours, it can be speculated that the CCNE1-driven HCC progression may depend on *CDK2*-expression levels at a late stage. Thus, we investigated the expression of *CCNE1* or *CDK2* in well-defined patient HCC subtypes (iClusters) according to demographic, pathologic and molecular features found by integrative multi-platform clustering [40]. *CCNE1* expression was upregulated in tumour samples of two of the three major HCC subtypes, characterised by a common signature of high proliferation, low differentiation, high tumour grade, vascular invasion, chromosomal instability and immune infiltration [40,41]. Consistent with observations from our murine study, markers of HCC stem cell differentiation (*AFP*, *THY1*), invasive processes (*ICAM1*, *VIM*), leukocyte recruitment (*CCR5*) and myofibroblast activity (*COL1A1*) were highly associated with *CCNE1* expression in HCC patients. Concomitantly, *CDK2* expression was not associated with any HCC subtype and marker selection. Finally, we validated our gene signature in the same human HCC cohort by proteomic database (TCPA) analysis of 218 cancer-relevant protein structures [23,24]. Here, CCNE1 protein expression was highly associated with markers of proliferation signalling (e.g., MYC, NRAS, AKT, ERBB2, MET, CDK1), DDR (MSH2, RAD51, BRCA2) and metastasis-related signalling (SRC, STAT3, SNAI1). Our data suggest that CCNE1 is a key regulator for human HCC progression independent of CDK2. A final piece of evidence was gained from a small cohort of HCC patients, in which we identified proliferating tumour tissue lacking CDK2 kinase activity above background levels. This demonstrates that human HCC progression can principally occur without CDK2 activity. Altogether, we could confirm the key findings from our mouse experiments in the available human HCC data.

Accordingly, our study provides important new knowledge on the role of CCNE1 and CDK2 for HCC progression with important clinical implications. CCNE1, but not CDK2, could serve as a potent molecular diagnostic and prognostic biomarker in HCC, independent of patients’ risk factor history. Hence, pharmacological CDK2 inhibitors (which are already available) might not be ideal candidates for HCC therapy, while direct targeting of CCNE1 could be a very promising approach by disrupting CDK2-dependent and independent oncogenic functions. In this context, we recently showed that therapeutic targeting of *Ccne1* in vivo using RNA interference is technically possible and capable of halting the progression of liver fibrosis [7]. Future studies could expand this strategy to HCC therapy. The potency of therapeutic *CCNE1* inhibition on parenchymal tumour cell growth, stemness and invasion capacity should therefore be first addressed in 3D culture systems such as tumour spheres, hydrogel microspheres or organoids of hepatoma cells or patient-derived xenograft. These approaches allow inclusion and further investigation of the contribution of CCNE1 in cells of the microenvironment, such as fibroblasts and immune cells, for their contributions to HCC progression [46,47,48]. In addition, CCNE1 inhibition was shown to sensitise hepatoma cells to regorafenib [49]. Thus, combined treatment of HCC with *CCNE1*-siRNA and regorafenib or related drugs could be a future therapeutic concept in preclinical studies.

## 5. Conclusions

Our data suggest that overexpression of CCNE1 is a key driver of hepatocarcinogenesis by promoting several oncogenic events besides hepatoma cell proliferation in a CDK2-independent manner. Therapeutic inhibition of CCNE1 in the course of liver cancer progression significantly reduces tumour burden and should therefore be considered as a beneficial treatment option against HCC.

## Figures and Tables

**Figure 1 cancers-13-05680-f001:**
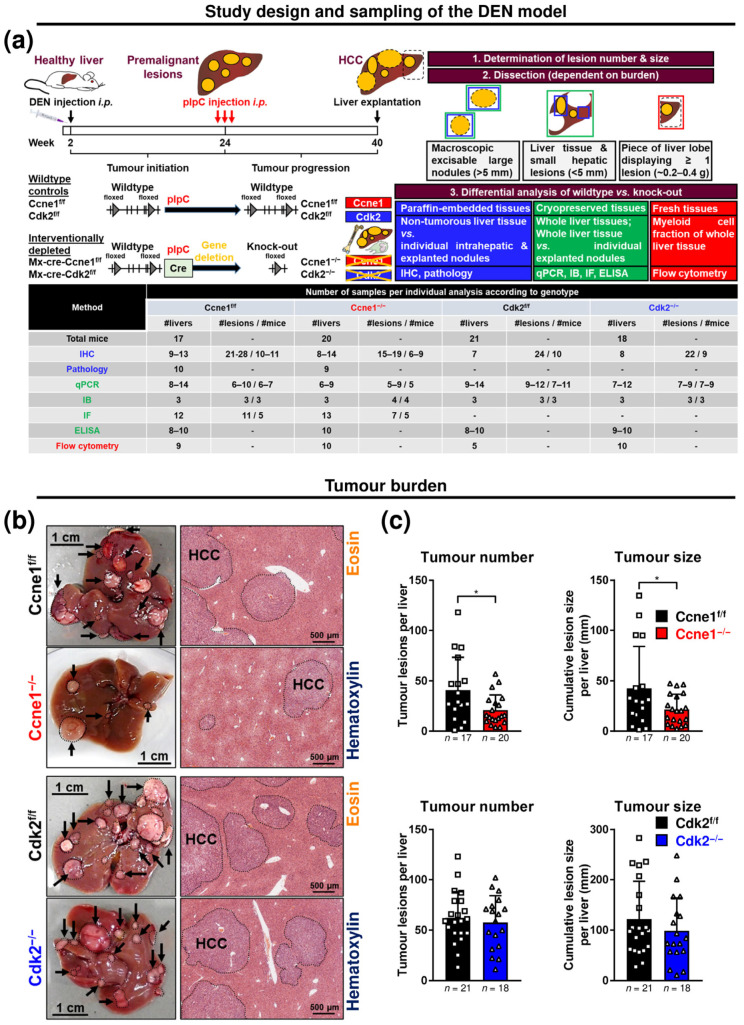
Diethylnitrosamine (DEN)-induced hepatocellular carcinoma (HCC) progression is driven by Cyclin E1 (CCNE1) independent of Cyclin-dependent kinase 2 (CDK2). (**a**) Experimental setting of the interventional HCC model. Top: *Mx-Cre*-transgenic mice and *Mx-Cre*-negative littermates with loxP-flanked (floxed, f/f) *Ccne1* or *Cdk2* allele were once treated intraperitoneally (i.p.) with the hepatocarcinogen DEN at the age of two weeks. After 22 weeks (tumour initiation), mice developed macroscopically visible premalignant liver lesions, which progressed to solid HCC until week 40 (tumour progression). All mice received three injections of poly-I:poly-C (pIpC) at the age of 24 weeks to induce the expression of the *Cre*-recombinase in liver cells, leukocytes and the hematopoietic stem cell compartment. This leads to the interventional gene deletion (−/−) of *Ccne1* or *Cdk2* specifically in *Mx*-*Cre*-positive mice through excisions of exons 5–11 in *Ccne1* or exon 2–3 in the *Cdk2 loci*. These mice were referred to as Ccne1^−/−^ and Cdk2^−/−^ mice (also see Appendix A). In contrast, pIpC treatment in *Mx*-*Cre*-negative mice does not affect the *Ccne1* or *Cdk2 loci* therefore serving as wildtype controls. Respective control mice were referred to as Ccne1^f/f^ and Cdk2^f/f^ (also see Appendix A). HCC progression in wildtype versus *Ccne1*/*Cdk2*-depleted animals was analysed after week 40. Bottom: table illustrating the number of collected samples for each individual analysis according to each genotype. Shown are number of livers (#livers), number of lesions taken from number of individual mice (#lesions/#mice). IHC, immunohistochemistry; pathology, investigation by an experienced pathologist (see Appendix A); IB, immunoblot analysis; IF, analysis by immunofluorescence microscopy; ELISA, determination of immunological mediators by enzyme linked immunosorbent assay; flow cytometry, analysis of liver tissue by fluorescence activated cell sorting. (**b**) Macroscopic appearance of representative livers and hematoxylin and eosin (H&E)-stained liver sections of Ccne1^f/f^ and Ccne1^−/−^ (top) or Cdk2^f/f^ and Cdk2^−/−^ mice (bottom). Arrows indicate dysplastic lesions; dotted lines: tumour areas. (**c**) Number of macroscopic HCC nodules and the cumulative liver tumour diameter (size, mm) for each Ccne1^f/f^ and Ccne1^−/−^ (top) or Cdk2^f/f^ and Cdk2^−/−^ (bottom) animal. Data are expressed as mean ± SD. *: *p* < 0.05.

**Figure 2 cancers-13-05680-f002:**
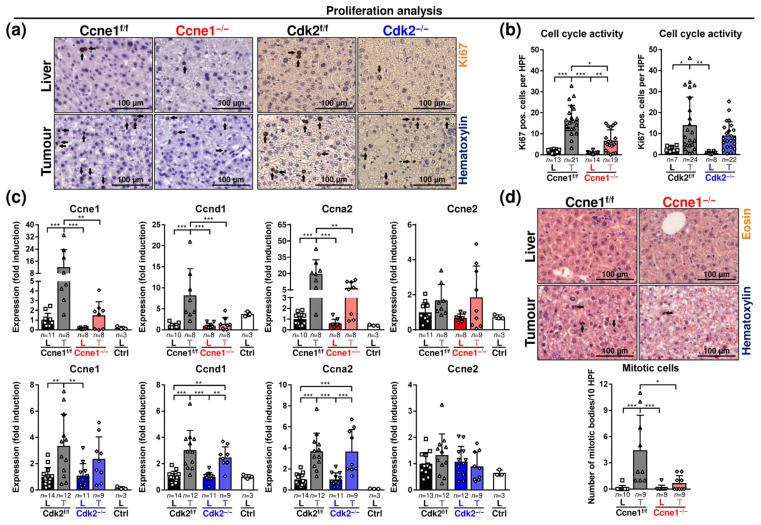
Depletion of *Ccne1* during HCC progression limits cell cycle activity. Two-week-old, *Mx-Cre* transgenic mice with floxed *Ccne1* or *Cdk2* alleles were treated with DEN. After tumour initiation, mice were repetitively injected with pIpC at week 22 to induce the deletion of *Ccne1* or *Cdk2* (Ccne1^−/−^, Cdk2^−/−^). *Mx*-*Cre*-negative mice also received both DEN and pIpC and served as wildtype controls (Ccne1^f/f^, Cdk2^f/f^). HCC progression was analysed after week 40. Details are given in Figure 1 and in the method section. L: liver; T: tumour tissue. (**a**) Immunohistochemical staining for Ki67 (brown) in paraffin sections of Ccne1^f/f^ and Ccne1^−/−^ (left) or Cdk2^f/f^ and CDK2^−/−^ (right) animals. Total nuclei were stained with haematoxylin (blue). Arrows indicate Ki67-positive cells; (**b**) quantification of Ki67-positive cells in non-tumorous liver (L) and tumorous tissues (T) of Ccne1^f/f^ and Ccne1^−/−^ (left) or Cdk2^f/f^ and CDK2^−/−^ (right) animals; (**c**) gene expression analysis of *Ccne1*, Cyclin D1 (*Ccnd1*), Cyclin A2 (*Ccna2*) and Cyclin E2 (*Ccne2*) in Ccne1^f/f^ and Ccne1^−/−^ (top) mice or Cdk2^f/f^ and Cdk2^−/−^ (bottom) animals. Ctrl: livers of untreated wildtype mice. Normalised expression levels were calculated as fold induction in comparison to DEN-treated control livers. (**d**) Top: mitotic activity in H&E-stained paraffin sections of non-tumorous liver tissues and tumorous tissues of Ccne1^f/f^ and Ccne1^−/−^ animals. Arrows indicate mitotic bodies. Bottom: quantification of number of mitotic bodies per 10 HPF at a magnification of 200x. HPF: high power field. Sample sizes are indicated within diagrams. Data are expressed as mean ± SD. *: *p* < 0.05; **: *p* < 0.005; ***: *p* < 0.001.

**Figure 3 cancers-13-05680-f003:**
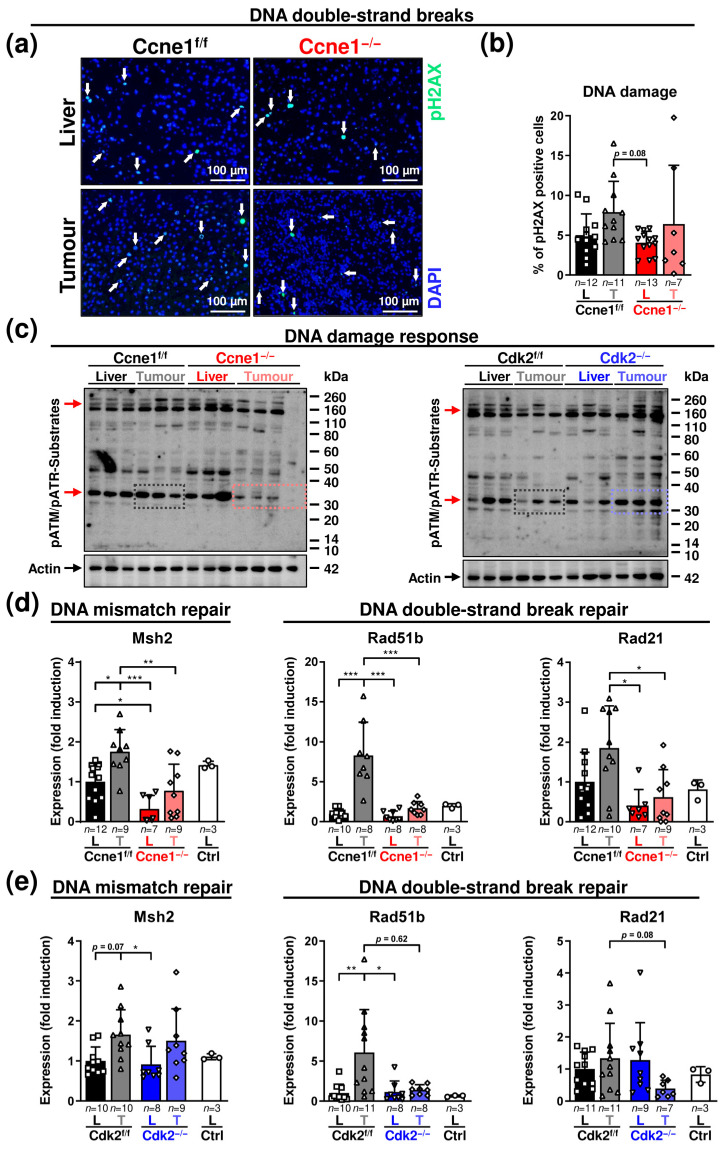
Absence of *Ccne1* in advanced HCCs modifies the response to DNA damage. Two-week-old, *Mx-Cre* transgenic mice with floxed *Ccne1* or *Cdk2* alleles were treated with DEN. After tumour initiation, mice were repetitively injected with pIpC at week 22 to induce the deletion of *Ccne1* or *Cdk2* (Ccne1^−/−^, Cdk2^−/−^). *Mx*-*Cre*-negative mice also received both DEN and pIpC and served as wildtype controls (Ccne1^f/f^, Cdk2^f/f^). HCC progression was analysed after week 40. Details are given in Figure 1 and in the method section. L: liver; T: tumour tissue. (**a**) Immunofluorescence staining of phosphorylated Histone 2AX (pH2AX, green) in liver and tumour cryosections. Total nuclei were stained with 4′,6-diamidino-2-phenylindole (DAPI, blue). Arrows indicate pH2AX-positive cells. (**b**) Quantification of pH2AX-positive cells (%). (**c**) Immunoblot analysis detecting phosphorylated substrates of Ataxia telangiectasia mutated/Ataxia telangiectasia and Rad3-related (ATM/ATR) kinases. Beta actin served as loading control. Red arrows: reduced phosphorylation of unknown substrates specifically in tumorous tissues of Ccne1^−/−^ animals. (**d**,**e**) Gene expression analysis of Mut-S homologue 2 (*Msh2*), *Rad51b* and *Rad21* in (**d**) CcnE1^f/f^ and CcnE1^−/−^ or (**e**) Cdk2^f/f^ and Cdk2^−/−^ animals. Ctrl: livers of untreated wildtype mice. Normalised expression levels were calculated as fold induction in comparison to DEN-treated control livers. Corresponding signalling pathways of the target genes are indicated in the headings. Sample sizes are indicated within diagrams. Data are expressed as mean ± SD. *: *p* < 0.05; **: *p* < 0.005; ***: *p* < 0.001.

**Figure 4 cancers-13-05680-f004:**
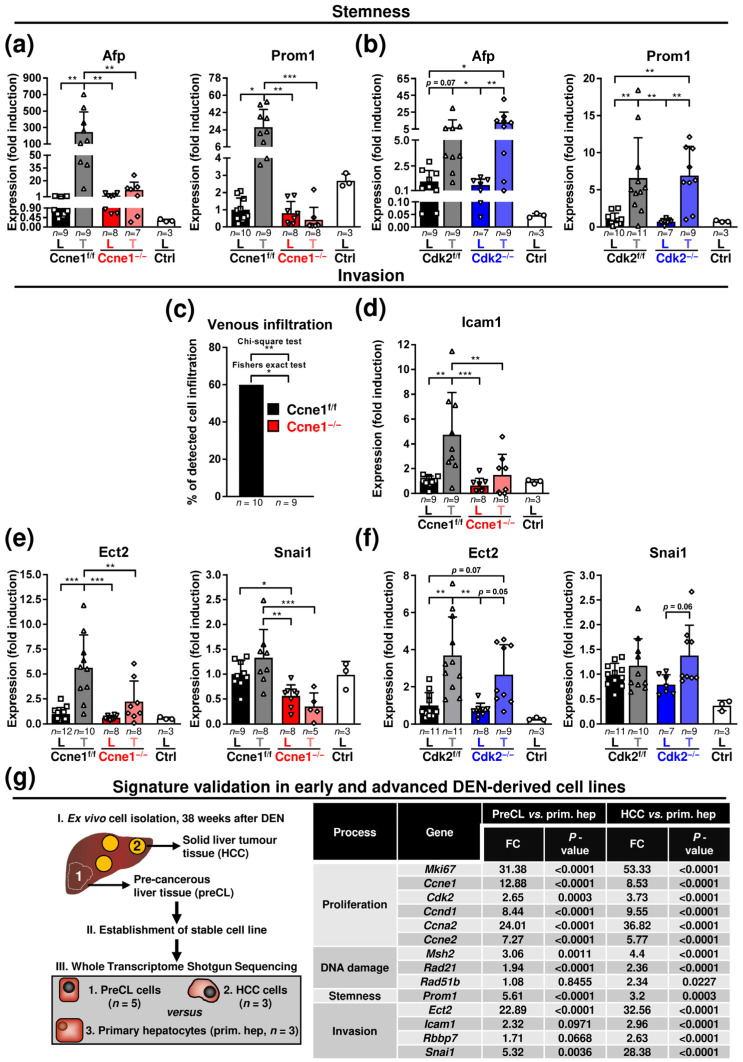
HCC progression after interventional deletion of *Ccne1* is characterised by diminished stemness and microinvasion. Two-week-old *Mx*-*Cre* transgenic mice with floxed *Ccne1* or *Cdk2* alleles were treated with DEN. After tumour initiation, mice were repetitively injected with pIpC at week 22 to induce the deletion of *Ccne1* or *Cdk2* (Ccne1^−/−^, Cdk2^−/−^). *Mx*-*Cre*-negative mice also received both DEN and pIpC and served as wildtype controls (Ccne1^f/f^, Cdk2^f/f^). HCC progression was analysed after week 40. Details are given in Figure 1 and in the method section. L: liver; T: tumour tissue. (**a**,**b**) Gene expression analysis of alpha fetoprotein (*Afp*) and Prominin 1 (*Prom1*) in (**a**) CcnE1^f/f^ and CcnE1^−/−^ or (**b**) Cdk2^f/f^ and Cdk2^−/−^ animals. Ctrl: livers of untreated wildtype mice. Normalised expression levels were calculated as fold induction in comparison to DEN-treated control livers. Data are expressed as mean ± SD. (**c**) Incidence (%) of venous infiltration (microinvasion, V1) in CcnE1^f/f^ and CcnE1^−/−^ mice. H&E sections (not shown) were analysed for presence of hepatoma cells in venous vessels by an experienced pathologist (N.G., see Appendix A). (**d**) Gene expression analysis of Intracellular adhesion molecule 1 (*Icam1*) in CcnE1^f/f^ and CcnE1^−/−^. (**e**,**f**) Epithelial cell transforming sequence 2 (*Ect2*) and snail family transcriptional repressor 1 (*Snai1*) in (**e**) CcnE1^f/f^ and CcnE1^−/−^ or (**f**) Cdk2^f/f^ and Cdk2^−/−^ animals. Ctrl: livers of untreated wildtype mice. Normalised expression levels were calculated as fold induction in comparison to DEN-treated control livers. Corresponding signalling pathways of the target genes are indicated in the headings. Sample sizes are indicated within diagrams. Data are expressed as mean ± SD. *: *p* < 0.05; **: *p* < 0.005; ***: *p* < 0.001. (**g**) In vitro validation of gene expression signature in early and advanced murine DEN-derived hepatoma cell lines (GEO database, accession no. GSE111079, accessed on 22 October 2021). Left: schematic illustration of sample generation. (I.) Primary murine hepatoma cells were ex vivo isolated from solid tumours (1) or surrounding liver (2) tissues of DEN-treated, Cre-negative C57B6 animals with a floxed allele of *Ccne1* or *Cdk2* at the age of 40 weeks and (II.) cultured to obtain stable cell lines. (III.) Gene expression levels of tumour-derived advanced HCC cells (1. HCC, *n* = 3) or precancerous liver cells (2. preCL, *n* = 5) were compared to non-transformed primary isolated hepatocytes (3. prim. Hep, *n* = 3) by Whole Transcriptome Shotgun Sequencing. Right: table of gene expression levels of preCL and HCC cells as fold change (FC) to primary hepatocytes for markers of proliferation (*Mki67*, *Ccne1*, *Cdk2, Ccnd1*, *Ccna2* and *Ccne2*), stemness (*Prom1*), DDR (*Msh2 Rad21* and *Rad51b*) and invasion (*Ect2*, *Rbbp7*, *Icam1* and *Snai1*). *p*-values are indicated for each gene.

**Figure 5 cancers-13-05680-f005:**
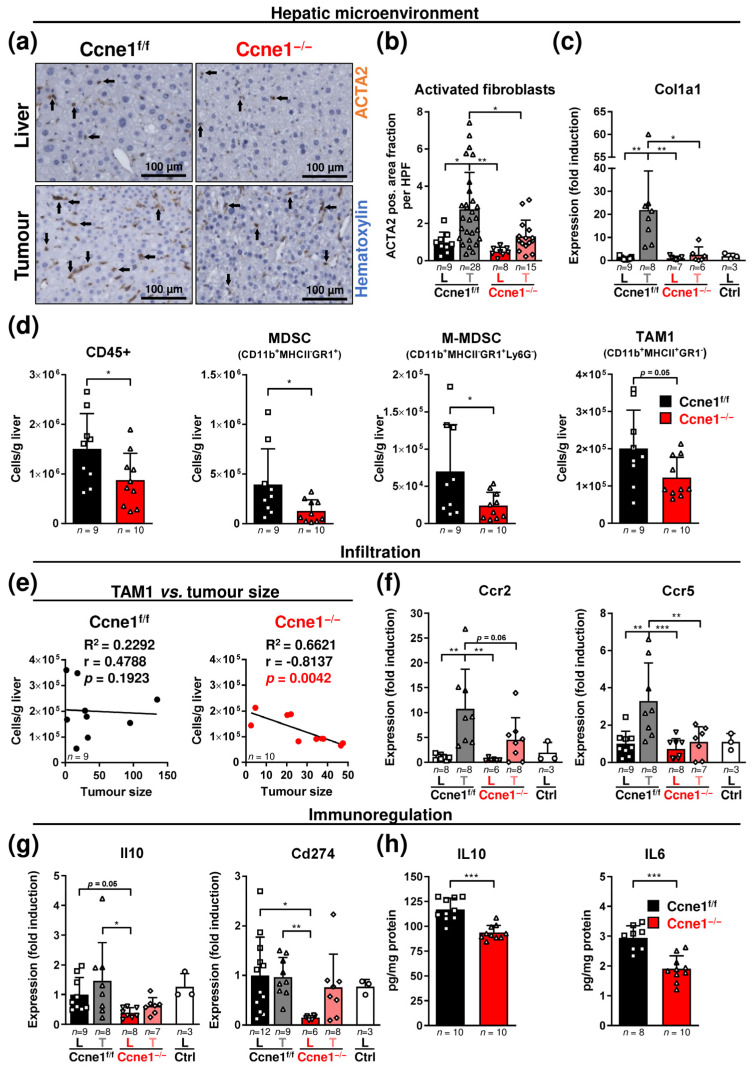
Interventional deletion of *Ccne1* is associated with reduced presence of activated myofibroblasts and myeloid cells during HCC progression. Two-week-old *Mx*-*Cre* transgenic mice with floxed *Ccne1* allele were treated with DEN. After tumour initiation, mice were repetitively injected with pIpC at week 22 to induce the deletion of *Ccne1* (Ccne1^−/−^). *Mx*-*Cre*-negative mice also received both DEN and pIpC and served as wildtype controls (Ccne1^f/f^). HCC progression was analysed after week 40. Details are given in Figure 1 and in the method section. L: liver; T: tumour tissue. (**a**) Immunohistochemical smooth muscle actin alpha 2 (ACTA2) staining (brown) of paraffin sections in tumour-free liver and tumorous tissues. Total nuclei were stained with hematoxylin (blue). Arrows indicate ACTA2-positive cells. (**b**) Quantification of ACTA2-positive area fraction at a magnification of 200×. HPF: High power field. (**c**) Gene expression analysis of collagen type I alpha-1 chain (*Col1a1*). Ctrl: livers of untreated wildtype mice. Normalised expression levels were calculated as fold induction in comparison to DEN-treated control livers. (**d**) Determination of immune cell populations according to surface marker expression, cluster of differentiation 45 (CD45)-positive cells, myeloid-derived suppressor cells (MDSC), monocytic MDSCs (M-MDSC), granulocytic MDSCs (G-MDSC), tumour-associated macrophages 1 (TAM1), as cells per gram liver by cytometry. The gating strategy is shown in Appendix A and described in the method section. Data are expressed as mean ± SD. (**e**) Linear analysis. Correlation between the cumulative tumour size and amount of TAM1. R^2^: Coefficient of determination, r: Pearson correlation coefficient, *p*-values are indicated within diagrams. (**f**,**g**) Gene expression analysis of (**f**) CC chemokine receptor 2 (*Ccr2*), CC chemokine receptor 5 (*Ccr5*), (**g**) interleukin 10 (*Il10*) and cluster of differentiation 274 (*Cd274*). Ctrl: livers of untreated wildtype mice. Normalised expression levels were calculated as fold induction in comparison to DEN-treated control livers. (**h**) Determination of IL10 and interleukin 6 (IL6) concentrations in DEN-treated livers by ELISA. Corresponding signalling pathways of the target genes are indicated in the headings. Sample sizes are indicated within diagrams. Data are expressed as mean ± SD. *: *p* < 0.05; **: *p* < 0.005; ***: *p* < 0.001.

**Figure 6 cancers-13-05680-f006:**
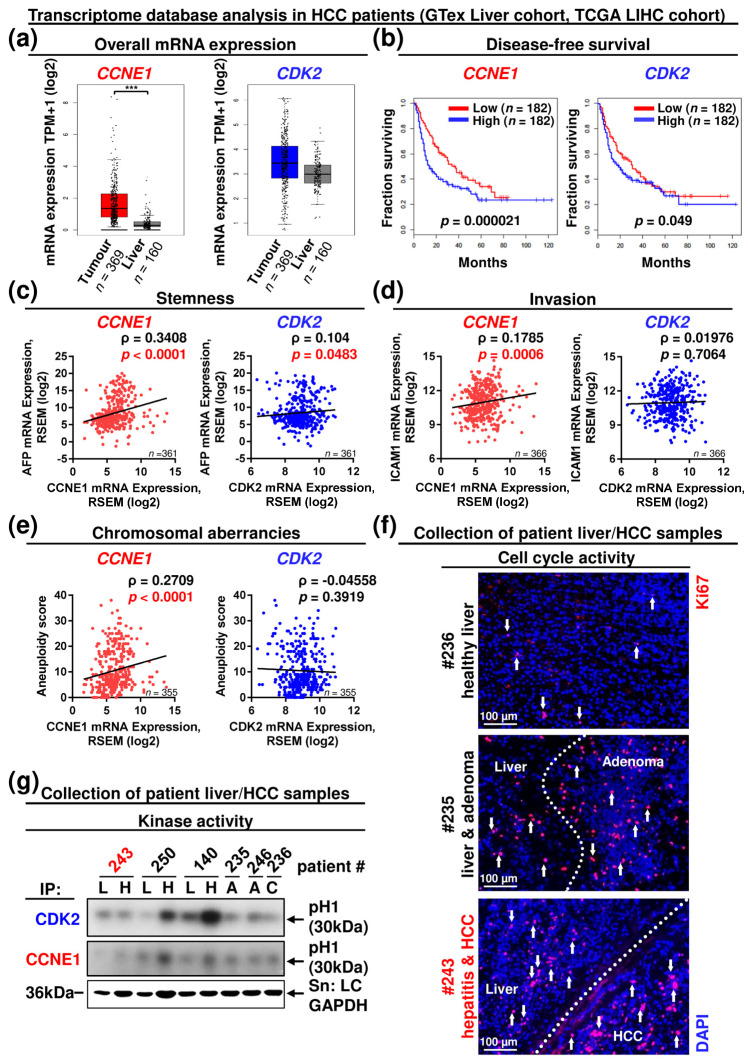
The *CCNE1* but not the *CDK2* level contributes to a gene signature of dedifferentiation, invasion and chromosomal instability. In silico database analysis of *CCNE1* and *CDK2* expression in tumour samples and matching tumour-free liver samples of HCC patients based on RNA-seq data generated by the TCGA Research Network, liver hepatocellular carcinoma (LIHC) cohort (http://cancergenome.nih.gov/, *n* = 373 HCC samples, *n* = 50 liver samples, accessed on 5 May 2021) and independent tumour-free liver samples (*n* = 110) by GTex liver cohort (http://gtexportal.org/, accessed on 5 May 2021). Details are given in the methods section. (**a**) mRNA expression levels of *CCNE1* (left) and *CDK2* (right) in HCC patients and healthy livers (according to http://gepia2.cancer-pku.cn/, accessed on 5 May 2021). TPM: Transcript per kilobase million. ***: *p* < 0.001. (**b**) Disease-free survival of HCC patient cohorts (according to http://gepia2.cancer-pku.cn/, accessed on 5 May 2021) with high or low expression of *CCNE1* (left) or *CDK2* (right). (**c**–**e**) Linear analysis in human HCC patients. (**c**) Correlation between mRNA expression levels of CCNE1 (left) or *CDK2* (right) and *AFP* mRNA expression. (**d**) Correlation between mRNA expression levels of *CCNE1* (left) or *CDK2* (right) and *ICAM1* mRNA expression. (**e**) Correlation between mRNA expression levels of *CCNE1* (left) or *CDK2* (right) and aneuploidy score (sum of altered chromosome arms). Corresponding signalling pathways of the target genes are indicated in the headings. ρ: Spearman correlation coefficient, *p*-values and sample sizes are indicated within diagrams. RSEM: RNA-Seq by Expectation–Maximisation. (**f**,**g**) Analysis of a small cohort of human hepatocellular carcinoma samples (H) and matching surrounding liver (L) tissues (#243, 250, 140), adenomas (A, #235, 246) and healthy control liver (C, #236). Patient details are given in Appendix A. (**f**) Immunofluorescence staining of Ki67 (red) in liver cryosections of selected patients displaying healthy control liver (#236), benign liver adenoma (#235) or HCC (#243). Dashed lines highlight the border between tumorous and non-tumorous tissue. Total nuclei were stained with DAPI (blue). Arrows indicate Ki67-positive cells. (**g**) In vitro kinase assay from human liver and HCC samples after immunoprecipitation (IP) of CDK2 or CCNE1. Arrow: phosphorylation of the substrate histone H1 (pH1). GAPDH expression served as input control. Red: proliferative HCC sample lacking CDK2 kinase activity.

## Data Availability

Human transcriptomic data are based upon data generated by the TCGA Research Network (https://www.cancer.gov/tcga, accessed on 5 May 2021) and GTex (http://gtexportal.org/, accessed on 5 May 2021). Whole Transcriptome Shotgun Sequencing data of murine hepatoma cell lines have been deposited in the Gene Expression Omnibus (GEO) database, https://www.ncbi.nlm.nih.gov/geo/ (accession no. GSE111079, accessed on 22 October 2021).

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
