# Peer review of "Cyclin E1 in Murine and Human Liver Cancer: A Promising Target for Therapeutic Intervention during Tumour Progression"

_cancers, 2021, doi:10.3390/cancers13225680_

Round 1

Reviewer 1 Report

In this article, the author examed the contributions of CCNE1 and CDK2 for HCC progression in both mice and humans. They provide sound evidence to support their conclusion that CCNE1 is a key driver of hepatocarcinogenesis. Overexpression of CCNE1 promotes several oncogenic events besides hepatoma cell proliferation which is independent of CDK2 expression. The article is able to be accepted in the present form.

Reviewer 2 Report

In the present manuscript, the authors discuss on specific inhibition of cyclin E1 as a potent therapeutic strategy for liver cancer patients.

I have several reservations. My comments are appended as below:

Major comments:

  1. Introduction – lines 58-59: please mention the prognosis period.
  2. qPCR- please explain in detail, how RNA quantity was determined, which kit was used for cDNA preparation?
  3. How non-tumourous liver tissues were defined? (line 113)
  4. Figure 1C- what was the cutoff size to calculate tumor no? Authors should see the median/mean tumor diameter among different groups.
  5. I am curious to know which pathological type the induced HCC in animals corresponds to that observed in humans?
  6. Figure 4- do authors check the expression of other stem cell-associated genes as OCT/Sox/Nanog?
  7. In addition to animal models, authors should perturb the CDK/CCNE gene expression in cell lines and check the stem cell, invasion markers.
  8. In animal models, authors should do the sphere assay to validate the stem cell involvement.
  9. Figure 6- which pathological grade does the tumor tissue belongs?
  10. Do authors think the CDK/CCNE perturbation influences the EMT process in addition to what authors observe?
  11. Figure 6b- how authors stratify samples?
  12. It is not clear to me on the therapeutic front how do authors propose to intervene in the CCNE expression. The authors should explain.

Minor comments:

  1. IHC and western and IP- please indicate the primary and secondary antibody dilution.
  2. Authors should provide catalog number of all used reagents/kits and constructs.

Reviewer 3 Report

This paper is well written and helpful for researchers on cancers.

As a comment, the recent reports on in vitro culture based on the relationship between tumor microenvironment and liver cancer cells should be introduced in the Discussion. Because the authors' paper must affect these fields in the future.

I suggest recent papers be added.

Research papers

Tissue Eng. Part A, 26, 2020, 1272-1282.

Biomaterials 159 (2018) 229-240

Review papers

Cancers 202012(10), 2754

Round 2

Reviewer 2 Report

I congratulate the authors for modifying the manuscript. I however suggest taking note of the following minor points:

  1. In the modified manuscript, I could not locate the figures incorporated. I suggest authors have a look at it.
  2. Response 8- I agree with the explanation provided by the authors. I suggest including a section on the limitations of this study in the late part of the discussion, so that it may prime the potential researchers to take on the continuous study.
  3. Section 2.8- please include the kit catalog number.
  4. Please include a hypothesis figure for better understanding.
